# Local Structure and Redox Properties of Amorphous CeO_2_-TiO_2_ Prepared Using the H_2_O_2_-Modified Sol-Gel Method

**DOI:** 10.3390/nano11082148

**Published:** 2021-08-23

**Authors:** Myungju Kim, Gwanhee Park, Heesoo Lee

**Affiliations:** 1Department of Materials Science and Engineering, Pusan National University, Busan 46241, Korea; myungjukim@pusan.ac.kr; 2Graduate School of Convergence Science, Pusan National University, Busan 46241, Korea; gwanheepark@pusan.ac.kr

**Keywords:** CeO_2_-TiO_2_ nanoparticle, metal–oxygen linkage, peroxy ions, disordered oxygen, redox property

## Abstract

Amorphous CeO_2_-TiO_2_ nanoparticles synthesized by the H_2_O_2_-modified sol-gel method were investigated in terms of the Ce-*O*-Ce and Ti-*O*-Ti linkage, local structure, and redox properties. The decrease in the crystallinity of CeO_2_-TiO_2_ by H_2_O_2_ addition was confirmed. The metal–oxygen linkage analysis showed the difference in size of the metal–oxygen network between crystalline CeO_2_-TiO_2_ and amorphous CeO_2_-TiO_2_ due to the O_2_^2−^ formed by H_2_O_2_. The local structure of CeO_2_-TiO_2_ was analyzed with an extended X-ray absorption fine structure (EXAFS), and the oscillation changes in the k space revealed the disordering of CeO_2_-TiO_2_. The decrease in Ce-O bond length and the Ce-O peak broadening was attributed to O_2_^2−^ interfering with the formation of the extended metal–oxygen network. The temperature-programmed reduction of the H_2_ profile of amorphous CeO_2_-TiO_2_ exhibited the disappearance of the bulk oxygen reduction peak and a low-temperature shift of the surface oxygen reduction peak. The H_2_ consumption increased compared to crystalline CeO_2_-TiO_2_, which indicated the improvement of redox properties by amorphization.

## 1. Introduction

Metal oxides have been widely studied in from a number of research angles, including their uses in catalysts, in electrochemistry, and as gas sensors [1]. Cerium oxide, CeO_2_, is an important rare-earth oxide with a redox pair of Ce^3+^/Ce^4+^, which enables the storage/release of oxygen located at the tetrahedral site under an oxidation/reduction atmosphere [2]. However, CeO_2_ is mainly used with other metal oxides because of its low redox property, specific surface area, and the thermal stability of pure CeO_2_. TiO_2_ has been extensively investigated as a CeO_2_-TiO_2_ mixed oxide due to its advantages, such as a highly specific surface area, outstanding thermal and chemical stability, and nontoxicity. CeO_2_-TiO_2_ nanoparticles have attracted much interest for use in various areas where oxidation/reduction reactions of cerium ions are important, such as de-NOx catalysts, photocatalysts, and water gas shift catalysts [3,4,5]. CeO_2_-TiO_2_ nanoparticles have various properties depending on their crystallinity, and many studies have reported the improvement of the catalytic performance of CeO_2_ and TiO_2_ through amorphization [6,7].

Amorphous oxides have great potential, and their method of synthesis has been highlighted, as amorphous oxides have excellent physicochemical properties compared to crystalline oxides [8]. CeO_2_-TiO_2_ nanoparticles are synthesized by the sol-gel method, hydrothermal routes, and coprecipitation [9,10,11]. The sol-gel method is widely used and has the advantages of obtaining uniform particles and a highly specific area [12]. The structure of the oxide synthesized through this method is modified according to the degree of hydrolysis, which is affected by additives or the amount of H_2_O [13,14]. Modifying the hydrolysis and condensation reaction causes changes in structure—for example, in the crystallinity and microstructure [15,16,17]. Therefore, it is important to understand the mechanism of hydrolysis and condensation reactions in terms of local structure in the sol-gel reaction and to reveal the relationship between structural properties and physicochemical properties.

In this study, the metal–oxygen linkage, local structure, and redox properties of CeO_2_-TiO_2_ nanoparticles were synthesized by the H_2_O_2_-modified sol-gel method. The effect of H_2_O_2_ on the linkage in the sol-gel was verified by Fourier transform infrared spectroscopy. The local structure was analyzed using an extended X-ray absorption fine structure (EXAFS). The relationship between the local structure and the redox properties was examined by H_2_-temperature-programmed reduction.

## 2. Experimental Procedure

### 2.1. Sample Preparation

The samples were synthesized by the sol-gel method with a molar ratio of Ce:Ti = 3:7; cerium nitrate and titanium isopropoxide (TTIP, Ti[OCH(CH_3_)_2_]_4_, Sigma-Aldrich, St. Louis, MO, USA, 97%) were used as precursors of CeO_2_ and TiO_2_, respectively. TTIP was added to ethanol while stirring. Here, we added H_2_O_2_ at a volume ratio of TTIP:35% H_2_O_2_ = 1:5 after the addition of TTIP. Distilled water was poured into the mixed solution to start the hydrolysis of TTIP. Cerium nitrate hexahydrate (Ce(NO_3_)_3_ 6H_2_O, Sigma-Aldrich, St. Louis, USA, MO, 99%) was added to the mixed solution, then ammonia hydroxide solution (NH_4_OH 25% in H_2_O, Junsei, Tokyo, Japan) was slowly added until the pH of the mixed solution reached approximately 10. The filtered precipitate was dried at 80 °C and calcined at 550 °C to obtain a CeO_2_-TiO_2_ powder. The prepared oxides were named CT and CT_H_2_O_2_ according to the H_2_O_2_ addition.

### 2.2. Characterization

Powder X-ray diffraction (XRD) was measured by an X-ray diffractometer (X’pert pro MPD, Almelo, The Netherlands) with Cu Kα (40 kV, 40 mA) radiation. High-resolution transmission electron microscopy (HR-TEM) images and selected-area electron diffraction (SAED) patterns were obtained using field emission (JEOL JEM-2100, Tokyo, Japan) at the KBSi Busan Center. Fourier transform infrared spectroscopy (FTIR) spectra were recorded using a Vertex 80 v FTIR spectrometer (Bruker, Billericay, MA, USA) over the range of 400~4000 cm^−1^ with KBr pellets.

X-ray absorption spectroscopy experiments were conducted at the Ce *L*_3_-edge using the extended X-ray absorption fine structure (EXAFS) facility of the 10C-wide XAFS beam line at the Pohang Accelerator Laboratory (Pohang, Korea). The storage ring was operated at 2.5 GeV with an injection current of 251 mA using a Si (111) double-crystal monochromator. The X-ray absorption spectroscopy (XAFS) data at the Ce *L*_3_ edge were collected at room temperature in transmission mode. The background intensity was removed and normalized using Fourier transform under k-weight 3 to measure the interatomic distance. The XAFS data were analyzed using the Athena and Artemis software packages.

The Brunauer−Emmett−Teller (BET) surface area was measured by N_2_ adsorption/desorption with a Micromeritics 2020 M instrument (Micromeritics, Norcross, GA, USA). H_2_ temperature-programmed reduction (H_2_-TPR) was performed using a Quantachrome with a Chem BET chemisorption analyzer (Micromeritics, Norcross, GA, USA) with 100 mg of the sample in a quartz tube reactor with a thermal conductivity detector (TCD). The sample was preheated from room temperature to 200 °C for 1 h and then cooled down to room temperature. TPR was performed at 10 °C/min up to 800 °C using 10 vol% H_2_ in He gas. CuO was used as a reference for calibration to quantify the total amount of H_2_ consumed.

## 3. Results and Discussion

The crystallinity and phase of the synthesized CeO_2_-TiO_2_ were investigated by XRD analysis, as shown in Figure 1, and each pattern corresponded to CT and CT_H_2_O_2_. Anatase TiO_2_ and CeO_2_ peaks were confirmed in the CT pattern, revealing that CT consisted of CeO_2_ and TiO_2_ and that there were no secondary peaks. On the other hand, the broadening of those peaks was observed in the pattern of CT_H_2_O_2_ compared to CT, indicating that CT_H_2_O_2_ had a low crystallinity of CeO_2_ and TiO_2_ or consisted of very small nanoparticles. This finding is similar to that of previous studies, indicating that CT_H_2_O_2_ had very low crystallinity through a broad peak and a low intensity [18,19]. Further analysis was performed to examine the structure of the CeO_2_-TiO_2_ changed by H_2_O_2_.

The microstructure and crystallinity of CT and CT_H_2_O_2_ were verified through TEM images and SAED patterns, as displayed in Figure 2. Particles of approximately 10 nm with lattice fringes were observed in the TEM image of CT, indicating that they had a crystalline structure (Figure 2a). These crystalline structures were composed of a CeO_2_ (111) plane with an interplanar spacing of 0.31 nm and a TiO_2_ (101) plane with a 0.35 nm interplanar spacing. The SAED pattern of CT showed ring patterns, meaning that CT existed in crystalline form (Figure 2b). In contrast, an amorphous structure was found in the TEM image of CT_H_2_O_2_ (Figure 2c), and a diffuse ring pattern was confirmed (Figure 2d). Based on these results, H_2_O_2_ addition was the cause of the amorphization of CeO_2_-TiO_2_.

FTIR analysis was carried out to investigate the effect of H_2_O_2_ on CeO_2_-TiO_2_ linkage formation in sol-gel chemistry. The calcination process is generally necessary in the sol-gel synthesis of oxides, and impurities—including OH groups and organic matter—are removed when calcination is performed for precipitation [20]. The FTIR spectra of CT and CT_H_2_O_2_ before and after calcination are shown in Figure 3, and all peaks were classified through the literature [21,22,23,24,25]. The characteristic absorption peak of TTIP was in the range of 1085–1050 cm^−1^. Since characteristic peaks were not detected for all samples, all TTIPs participated in the sol-gel reaction. A reduced CH_3_ peak at 1338 cm^−1^ revealed that the organic matter was removed by calcination. Peaks assigned to the -OH group appeared at 3542 cm^−1^ and 1630 cm^−1^, and the peak intensities of the samples before calcination were greater than those after calcination. It should be noted that CT_H_2_O_2_ had stronger OH peaks than CT and had an O–O peak representing a peroxy group at 900 cm^−1^. In addition, the Ce-O_2_^2−^ peak at 842 cm^−1^ demonstrated that H_2_O_2_ formed O_2_^2−^ [26]. The difference between the OH peaks of CT and CT_H_2_O_2_ is explained by O_2_^2−^ (peroxy ions).

The Ce-*O*-Ce and Ti-*O*-Ti peaks in the fingerprint region of 800 to 400 cm^−1^ indicate the metal–oxygen linkage and bonding network. The one large peak in this region is due to the Ce-*O*-Ti linkage, confirming that the cerium species react via a cross-link and covalently bond on the Ti-*O*-Ti [25,27]. A wide peak in this region in the CT spectrum indicated a large network [28]. The intensity of this peak decreased after adding H_2_O_2_, and the intensity of this peak indicates the size of the Ce-*O*-Ce or Ti-*O*-Ti network. This decrease occurred because O_2_^2−^ interfered with the hydrolysis and condensation reactions and thus affected the formation of the Ti-*O*-Ti network.

EXAFS is a powerful technique for analyzing the local structure of amorphous materials [26]. The EXAFS k_3_χ data and Fourier transform (F.T.) of Ce L_3_ edge are shown in Figure 4. It was confirmed that CT had a long-range order but that CT_H_2_O_2_ had a short-range order resulting from the broken order. As corresponding results, the oscillation differences between CT and CT_H_2_O_2_ were clearly seen in the red dotted circle of the k space at the Ce *L*_3_-edge (Figure 4a), and severe noise due to the disordered structure of the amorphous phase was observed. EXAFS F.T. data at the Ce *L*_3_ edge (Figure 4b) provide information on the bond length and the disordering around the Ce atoms. The first and third peaks are assigned to the 1st shell and 2nd shell, respectively, indicating the distances between the Ce-O and Ce-Ce atoms. The second peak is the Ce-Ti bond assigned to an interaction between Ce-Ti forming amorphous Ce [29]. In CT, the bonding length of Ce-O was 1.84 Å and that of Ce-Ce was 3.53 Å. For CT_H_2_O_2_, the bond length of Ce-O decreased to 1.68 Å, and no Ce-Ce peak was observed. Additionally, the broadening of the 1st shell peak was confirmed with the addition of H_2_O_2_, which is related to amorphization due to disordering, revealing the disordering of the oxygen around Ce [30]. The peak intensity ratio of the 1st shell and the 2nd shell (I2nd/I1st) indicates the degree of amorphousness, and this ratio decreased from 0.56 to 0.15, indicating amorphization by H_2_O_2_.

This disordering occurs because the strong nucleophilicity of O_2_^2^^−^ causes strong bonding with Ti ions, which interferes with the binding of OH (Figure 5). In more detail, this could be described as the hydrolysis reaction (Equations (1) and (2)) and condensation reaction (Equations (3) and (4)) in sol-gel chemistry [15,17].
Hydrolysis: Ti(O*i*Pr)_4_ + 2H_2_O → Ti(OH)_4_ + 4*i*PrOH,(1)
Ti(O*i*Pr)_4_ + H_2_O_2_ + 2H_2_O → Ti(OO)^2−^_2_ + 4*i*PrOH + 6H^+^,(2)
Condensation: -Ti-OH + HO-Ti → -Ti-*O*-Ti- + H_2_O,(3)
-Ti-OH + *i*Pr*O*-Ti- → -Ti-*O*-Ti- + *i*PrOH.(4)

TTIP (Ti(O*i*Pr)_4_) reacts with H_2_O and a hydrolysis reaction occurs to form -Ti-OH (Equation (1)). The condensation reaction also simultaneously occurs to release one H_2_O or *i*PrOH molecule, forming an extended network of Ti-O-Ti (Equations (3) and (4)). In the H_2_O_2_ addition, O_2_^2−^ is strongly bound to Ti ions, eventually inhibiting the binding to OH (Equation (2)). As a result, the hydrolysis and condensation reactions of TTIP would have partially occurred. It is concluded that the extended network of Ti-*O*-Ti was not formed due to partial hydrolysis by O_2_^2−^, leading to amorphization.

The redox properties of CeO_2_-TiO_2_ according to amorphization were analyzed by H_2_-TPR, and the results are illustrated in Figure 6. The H_2_ consumption and specific surface area were 1110 mmol/g and 103.0 m^2^/g for CT, and 1354 mmol/g and 113.1 m^2^/g for CT_H_2_O_2_, respectively as shown in Table 1. This is due to the smaller particle size of CT_H_2_O_2_ or the formation of dangling bonds due to the disordering of Ce-O [31,32,33]. These two facts suggest that the active surface area of CT_H_2_O_2_ is larger. CT had two peaks observed at 490 °C and 767 °C, which corresponded to the reduction from Ce^4+^ to Ce^3+^ by the surface oxygen of CeO_2_ and the lattice oxygen of CeO_2,_ respectively [34]. CT_H_2_O_2_ showed a peak due to the surface oxygen at only 468 °C and 558 °C, and there was no reduction peak assigned to the lattice CeO_2_, which implies that CT_H_2_O_2_ is amorphous [35,36]. The surface CeO_2_ peak shifted to low temperature by amorphization, and the H_2_ consumption increased, confirming the improvement of the redox property. The atoms of the crystalline oxide are tightly bonded by strong intermolecular forces, while the atoms of the amorphous oxide are loosely bonded. Consequently, the improvement of the redox property was because the Ce-*O*-Ce bond was weak and easily reduced by the disordered structure of CT_H_2_O_2_.

## 4. Conclusions

We studied the effect of H_2_O_2_ on the Ce-*O*-Ti linkage in the sol-gel reaction in terms of local structure for the first time. The decrease in the crystallinity of CeO_2_-TiO_2_ with the addition of H_2_O_2_ was verified through XRD peak broadening, and it was confirmed that the SAED pattern of CeO_2_-TiO_2_ became a diffuse pattern. CeO_2_ and crystalline TiO_2_ were observed for CT, whereas an amorphous structure was observed for CT_H_2_O_2_. In the FTIR spectrum of CT_H_2_O_2_, the O_2_^2−^ peak attributed to H_2_O_2_ was observed, and the broadening of the metal–oxygen peak showed that the size of the metal–oxygen network decreased due to O_2_^2−^. The local structure of the synthesized samples was analyzed using EXAFS. Severe noise and oscillation differences in the k space were observed by the disordered structure in CT_H_2_O_2_; furthermore, the decrease in the Ce-O bond length and the broadening of the Ce-O peak in R space were identified. It was found that O_2_^2−^ interfered with the formation of the metal–oxygen extended network. In the H_2_-TPR analysis to investigate the redox property according to the amorphization, a low-temperature shift in the surface CeO_2_ reduction peak and an increase in H_2_ consumption indicated an improvement in the redox property due to amorphization. The improved redox property of amorphous CeO_2_-TiO_2_ resulted from the metal–oxygen bond being weakened by disordered oxygen.

## Figures and Tables

**Figure 1 nanomaterials-11-02148-f001:**
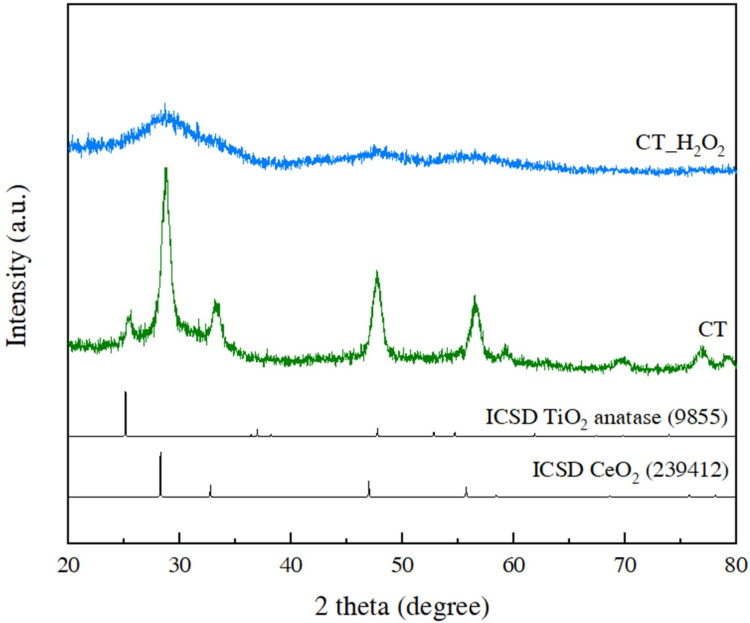
XRD patterns of CT and CT_H_2_O_2_.

**Figure 2 nanomaterials-11-02148-f002:**
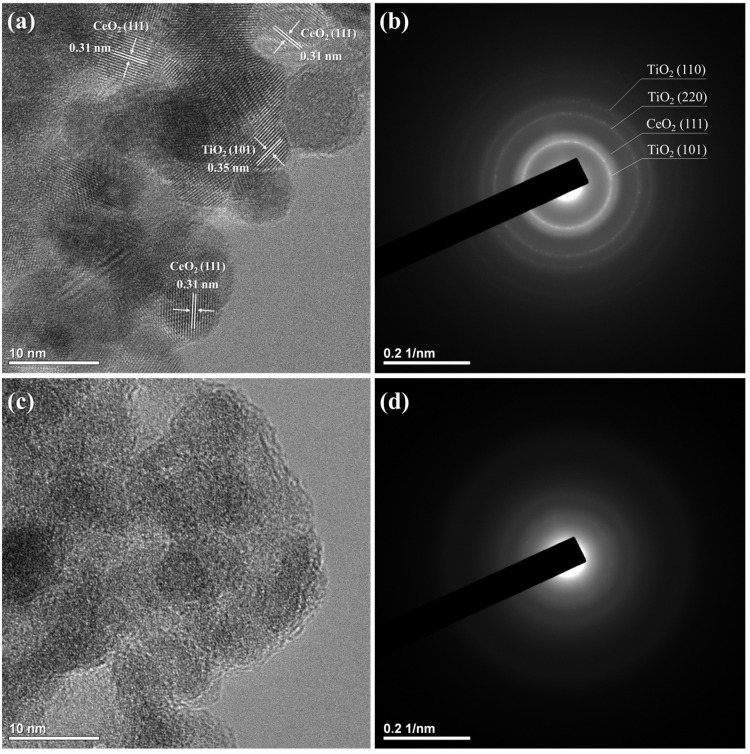
TEM images and SAED patterns of (**a**,**b**) CT and (**c**,**d**) CT_H_2_O_2_.

**Figure 3 nanomaterials-11-02148-f003:**
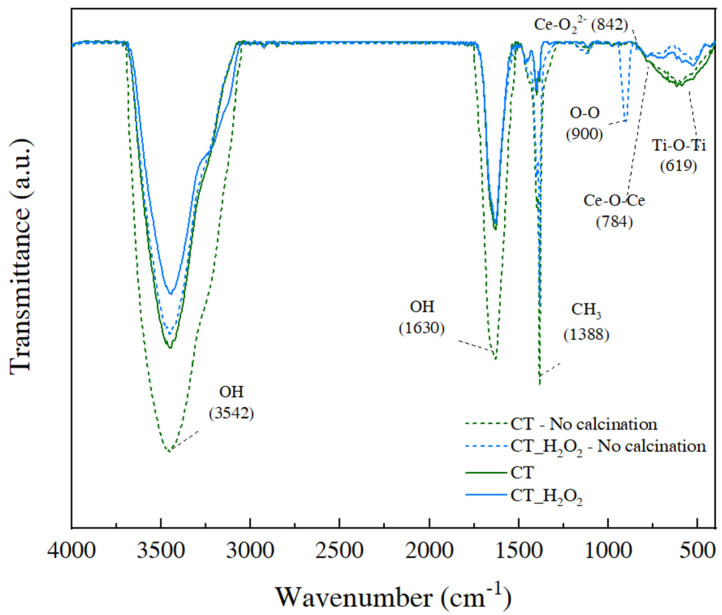
FTIR spectra of CT and CT_H_2_O_2_ before calcination.

**Figure 4 nanomaterials-11-02148-f004:**
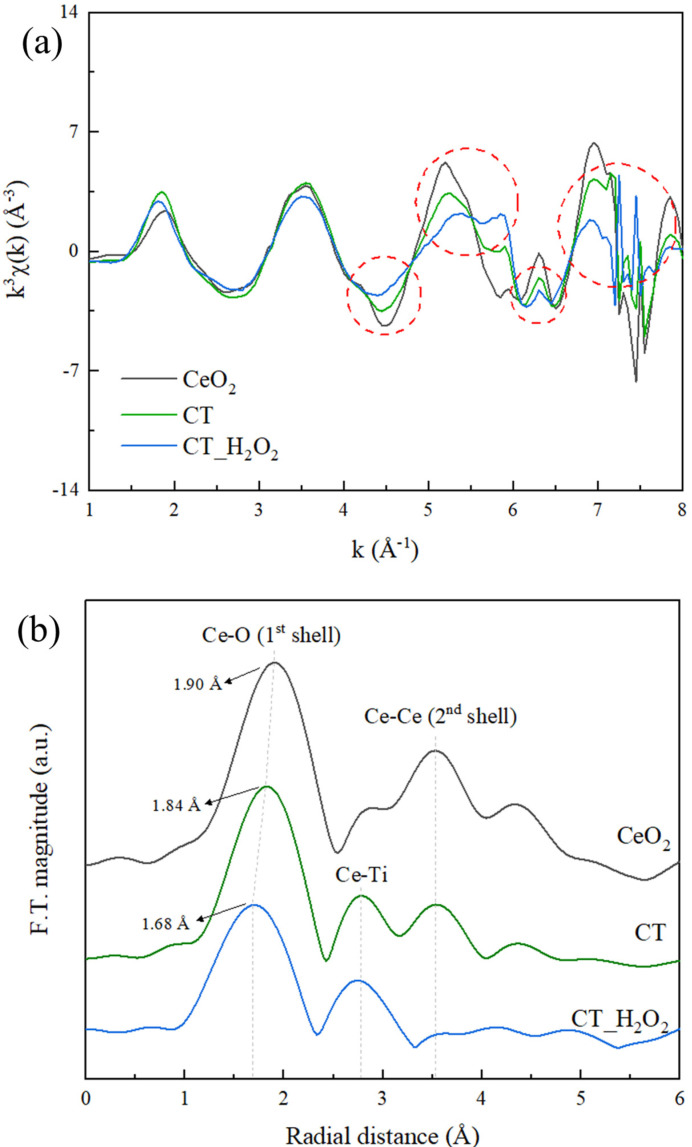
(**a**) Extended X-ray absorption fine structure (EXAFS) k_3_χ data and (**b**) Fourier transform of the EXAFS k_3_χ data at the Ce *L*_3_-edge EXAFS of CT and CT_H_2_O_2_.

**Figure 5 nanomaterials-11-02148-f005:**
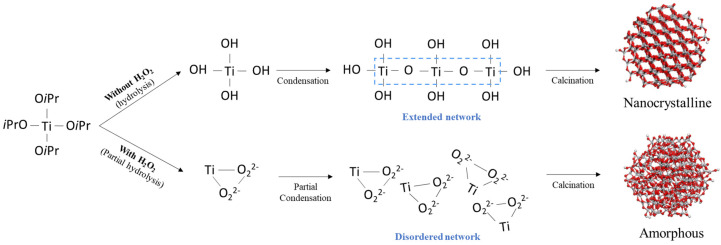
Schematic showing the effect of H_2_O_2_ on the local structure.

**Figure 6 nanomaterials-11-02148-f006:**
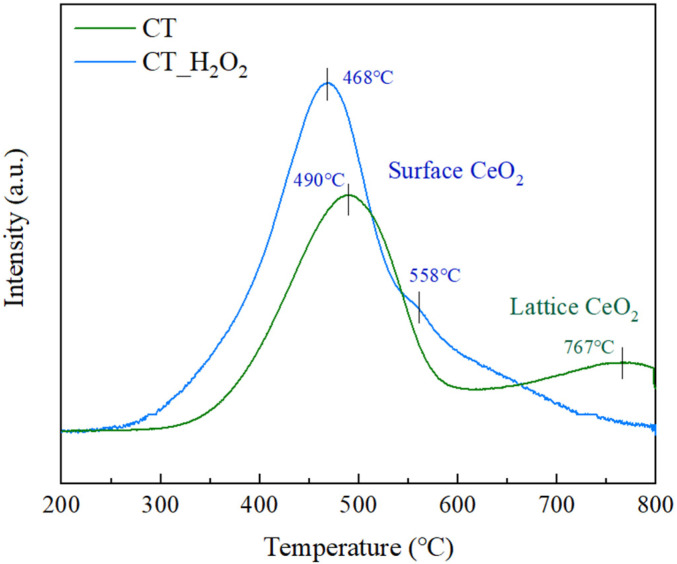
H_2_-TPR profiles of CT and CT_H_2_O_2_.

**Table 1 nanomaterials-11-02148-t001:** H_2_-TPR results and BET surface area of CT and CT_H_2_O_2_.

Sample	Surface Oxygen Reduction Temp. (°C)	Bulk Oxygen Reduction Temp. (°C)	H_2_ Consumption (mmol/g)	BET Surface Area (m^2^/g)
CT	490	797	1110	103.0
CT_H_2_O_2_	468	Not observed	1354	113.1

## Data Availability

The data presented in this study are available on request from the corresponding author.

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
