# Peer review of "Local Structure and Redox Properties of Amorphous CeO2-TiO2 Prepared Using the H2O2-Modified Sol-Gel Method"

_nanomaterials, 2021, doi:10.3390/nano11082148_

Round 1

Reviewer 1 Report

The paper authored by Kim and co-authors investigates crystalline to amorphous transition of CeO2-TiO2 system via the addition of H2O2 during the synthesis. This work is motivated by the increased activity of amorphous nanoparticles compared to crystalline counterparts. The paper is well written, although it contains mainly a description of experimental results with a rather limited discussion. The paper can be considered for publication in case the following points are addressed:

  • In the introduction section, the motivation for this work and the importance of the increased reactivity of amorphous nanoparticles compared to crystalline counterparts should be extended. While the crystalline-to-amorphous transition is the key topic of this paper, other aspects, such as the increased reactivity of nanoparticles with decreasing particle size (see, e.g., Fig. 2 in DOI:10.1038/NMAT2976) should be mentioned.
  • I find it surprising that the authors do not see TiO2(101) surface in SAED shown in Fig. 2b. The (101) surface is clearly visible in Fig. 1 and is also highlighted in Fig. 2a. If the (220) surface is the most prominent on TiO2, it should also be highlighted in Fig. 2a.
  • The CT_H2O2 system is shown to be more active than CT based on the H2-TPR results. Can you estimate the active surface area of the CT and CT_H2O2 system since the active surface area could be one of the key factors affecting the redox activity?
  • In section 2.2, line 73-74, states that the XAFS measurements were performed with a storage ring operated at 2.5 GeV with an injection current of 200 mA. If this is correct, then it would imply that the measurements were done more than ten years ago (The Pohang Light Source (PLS) was upgraded as the PLS-II in three years, 2009–2011. The electron beam energy was increased from 2.5 GeV to 3 GeV, and the beam current rose from 170 mA to 400 mA). Can you verify these experimental details?
  • On line 176, the authors mention that CT_H2O2 showed only one peak at 468 °C. There is a clear shoulder peak at 550 °C. The authors should comment on this. Also, it is very hard to see the bulk oxygen reduction for CT in Figure 6. It would help greatly to extend the horizontal axis to higher temperatures (e.g. 820 °C, so the bulk reduction of CT does not overlap with the border of the figure).
  • On line 115, the authors should consider replacing the word ‘chemistry’ with a more appropriate term since the peaks indicate the amount/concentration of the species rather than the chemistry of these samples.

Minor issues:

Line 72: the word 'Extended' is missing in the introduction of the EXAFS acronym

Author Response

We look forward to your positive response.

Thank you in advance.

Sincerely,

Heesoo Lee (corresponding author)

Reviewer 2 Report

  1. The introduction should be dedicated to a critical analysis of state-of-the-art related work to justify the objective of the study. In this case, overall, the introduction section is very poor, non-coherent and inadequate, without a comprehensive description.
  2. The authors did not elucidate clearly the novelty and motivation of the work. The main goal of the study was not clearly specified. Why these experiments were conducted? What did the Authors want to accomplish? In my opinion, answering to these questions will be helpful for readers to understand the data and its utility.
  3. More details regarding the modified sol-gel synthesis are necessary.
  4. Experimental procedure. The purity and grade of all chemicals, materials and solvents used in the study should be given under the materials sections. How was the crystallinity determined according to XRD?
  5. Results and discussion. To increase the scientific value of the manuscript, the authors should comment on the novelty of their approach and compare the results with similarly published papers.
  6. Please be sure that the Conclusions section not only summarize the key findings of your work but also explain the specific ways in which this work fundamentally advances the field relative to prior literature.

Finally, I believe that the work is not suitable for publication in this form and requires large revision. If the manuscript will not be considerable improved, I will not recommend its publication.

Author Response

(The authors gave the same response as above.)

Reviewer 3 Report

In this paper, the authors prepared amorphous CeO2-TiO2 nanoparticles by the H2O2-modified sol-gel.

For this paper, the answer is “Major Revision”. Please correct:

  • At Section “2.1. Sample preparation”, for the reagents indicate de manufacture, concentration (e.g., for titanium isopropoxide, cerium nitrate)
  • Please, specify the possible application.
  • Mention, why the H2O2-modified sol-gel method was used (the advantages in comparison with other method).
  • Relevant reference can be included:

„ Synthesis and morpho-structural properties of TiO2-based materials”, Journal of Optoelectronics and Advanced Materials 21(3-4), pp. 281-286

Author Response

(The authors gave the same response as above.)

Round 2

Reviewer 2 Report

Authors addressed referee's comments and improved their manuscript that is publishable in this form.

Reviewer 3 Report

Dear Sirs,

The authors made the recommended changes and the manuscript can be publish in this form.